# Neuromuscular Blockade in the Pre- and COVID-19 ARDS Patients

**DOI:** 10.3390/jpm12091538

**Published:** 2022-09-19

**Authors:** Vasiliki Tsolaki, George E. Zakynthinos, Maria-Eirini Papadonta, Fotini Bardaka, George Fotakopoulos, Ioannis Pantazopoulos, Demosthenes Makris, Epaminondas Zakynthinos

**Affiliations:** 1Critical Care Department, University Hospital of Larissa, Faculty of Medicine, University of Thessaly, 41110 Larissa, Greece; 2Third Department of Cardiology, Sotiria General Hospital, 11527 Athens, Greece; 3Neurosurgical Department, University Hospital of Larissa, Faculty of Medicine, University of Thessaly, 41110 Larissa, Greece; 4Emergency Department, University Hospital of Larissa, Faculty of Medicine, University of Thessaly, 41110 Larissa, Greece

**Keywords:** neuromuscular blocking agents, muscular relaxants, ARDS, survival, lung injury, COVID-19 ARDS

## Abstract

Acute respiratory distress syndrome (ARDS) accounts for a quarter of mechanically ventilated patients, while during the pandemic, it overwhelmed the capacity of intensive care units (ICUs). Lung protective ventilation (low tidal volume, positive-end expiratory pressure titrated to lung mechanics and oxygenation, permissive hypercapnia) is a non-pharmacological approach that is the gold standard of management. Among the pharmacological treatments, the use of neuromuscular blocking agents (NMBAs), although extensively studied, has not yet been well clarified. The rationale is to minimize the risk for lung damage progression, in the already-injured pulmonary parenchyma. By abolishing rigorous spontaneous efforts, NMBAs may decrease the generation of high transpulmonary pressures that could aggravate patients’ self-inflicted lung injury. Moreover, NMBAs can harmonize the patient–ventilator interaction. Recent randomized controlled trials reported contradictory results and changed the clinical practice in a bidirectional way. NMBAs have not been documented to improve long-term survival; thus, the current guidance suggests their use only in patients in whom a lung protective ventilation protocol cannot be applied, due to asynchrony or increased respiratory efforts. In the present review, we discuss the published data and additionally the clinical practice in the “war” conditions of the COVID-19 pandemic, concerning NMBA use in the management of patients with ARDS.

## 1. Introduction

The acute respiratory distress syndrome (ARDS) was first described in 1967 as a condition of respiratory failure that resembles the respiratory distress syndrome in infants [1]. The syndrome may originate from pulmonary (pneumonia, aspiration and chemical inhalational insults) or extra-pulmonary (trauma, burns, sepsis and pancreatitis) causes and it is acute in onset (within five days of the illness onset/insult). The inflammatory process results in increased vascular permeability (thus, a non-cardiac origin), leading to alveolar infiltration, increased lung weight and the loss of aerated lung tissue. Bilateral pulmonary infiltrates result in hypoxemia and decreased lung compliance [2]. 

Annually, ARDS accounts for 10% of intensive care unit (ICU) admissions and 24% of patients receiving mechanical ventilation [3]. Attributable mortality remains high, ranging from 35% to 46% and is associated with a degree of lung impairment [2,3]. Survivors may have significant impairments in their quality of life both regarding physical and neurocognitive functions, derangements that may persist for as long as 5 years after recovery from ARDS [4]. During the last three years, the novel coronavirus disease (COVID-19) has severely burdened healthcare system capacities in many parts of the world. A high proportion of patients require hospitalization, and a small subset will develop severe respiratory failure. Due to its pandemic nature (a substantial number of patients suffering at the same period), COVID-19 ARDS patients have overwhelmed ICUs [5,6,7]. Moreover, a high mortality has been reported in those patients receiving invasive mechanical ventilation (IMV), in the range of 47.9–84.4% [5,6,8]. 

Irrespective of the cause, a definite ARDS treatment is lacking. Its management mainly relies on supportive care, while lung-protective mechanical ventilation strategy is one of the major prerequisites. This includes the application of a low tidal volume on the mechanical ventilation and monitoring of inspiratory pressures, so that the plateau pressure does not exceed the value of 30 cm H_2_O and/or the driving pressure is kept below 14 cm H_2_O [9,10]. The application of higher positive-end expiratory pressure (PEEP), a strategy that has been adopted to minimize atelectrauma in recent decades [11], has been highly debated in ARDS management in the COVID-19 era [12,13,14], while proning became a routine clinical practice, and more than 60% of the patients are proned early after intubation [15]. Concerning pharmacological treatments, apart from the recently endorsed corticosteroids and IL-6 receptor antagonist, conferring a variable, clinically significant, survival benefit to COVID-19 patients [16,17,18], neuromuscular blocking agents (NMBAs) are also being used in the management of ARDS [3]. The rational for their use is to harmonize the patient–ventilator interaction, thus reducing the risk of progression to ventilator-induced lung injury (VILI), and a more homogenous distribution of pressurization during tidal ventilation [19]. The use of paralytics varies widely in everyday clinical practice [3]. Two large randomized clinical trials have conferred a bidirectional change in the clinical use of NMBAs in ARDS, as they report conflicting results mainly concerning mortality [20,21]. COVID-19 provided a “bolus” of ARDS patients. The aim of the present review is to present data concerning the role of muscle relaxants in the management of ARDS patients and, secondly, to delineate their use in the COVID-19 era. 

## 2. Mechanisms of Action, Pharmacology 

Pharmacologic paralysis or neuromuscular blockade refers to the pharmacological blocking of nerve impulses at the neuromuscular junction, resulting in skeletal muscle paralysis. Physiologically, acetylcholine (Ach) is released from the presynaptic motor nerve terminus, diffuses across the synaptic cleft, and binds to ligand-gated nicotinic acetylcholine receptors (nAchRs) on the postsynaptic motor endplate. Acetylcholine binds to its receptors, and this increases the membrane permeability to ions, thus decreasing transmembrane potential. Upon reaching the threshold potential, the action potential is propagated over the skeletal muscle cells, causing their contraction. The enzyme acetylcholinesterase rapidly terminates Ach action [22,23]. 

Neuromuscular blocking agents cause skeletal muscle relaxation as they block the Ach receptor at the neuromuscular junction [23]. There are two classes of NMBAs, depolarizing and non-depolarizing agents. The former are compounds that activate nAchRs, succinilcholine being the only agent in this category. It has a rapid onset and extremely short duration of action, ideal for a rapid sequence intubation but not suitable for continuous infusion. Its adverse effects are another issue that make the agent improper in the management of critically ill patients. Conditions resulting in the proliferation of extrajunctional AchRs (burns, immobilization, sepsis, muscle trauma and motor neuron lesions) may result in an augmented response in hyperkalemia, leading to fatal arrhythmias. It may also trigger malignant hyperthermia or an anaphylactic reaction. Other adverse effects include the increase in intracranial pressure, masseter spasm and cardiac dysrhythmias [22,24]. Non-depolarizing agents are highly ionized, water-soluble compounds that competitively antagonize nAchRs, preventing depolarization. They are further divided in aminosteroidal (rocuronium, vecuronium and pancuronium) and benzylisoquinolinium compounds (atracurium, cisatracurium and mivacurium). The choice between these agents depends on the indication and patients’ comorbidities [22]. Rocuronium, the most commonly used aminosteroidal agent, has a rapid onset and intermediate duration of action, making it suitable for rapid sequence intubation. Aminosteroidal compounds carry the risk for muscle weakness, due to their structure analogy; they carry a steroid moiety that may result in ICU weakness. Moreover, they present hepatic or renal metabolism and there is an increased risk of accumulation if used for several days [25]. There is limited experience with prolonged administration, although during the pandemic, due to drug shortages, rocuronium was also used for continuous infusion [26,27,28]. Adverse effects, such as central nervous system accumulation, have been reported, especially when the blood–brain barrier is affected [29,30]. On the other hand, benzylisoquinolines are metabolized via Hofmann elimination, a mechanism independent of organ function; they are inactivated by plasma esterases, their action depending on the plasma pH and temperature. Atracurium and cisatracurium are the preferred agents for continuous infusion, and, as the latter is not associated with histamine release, it is the NMBA of choice in the management of critically ill patients when myorelaxants are needed [26]. 

Factors affecting NMBAs’ pharmamokinetics and pharmacodynamics are concomitant drug use (carbamazepine, phenytoin and theophylline, imposing resistance to NMBAs’ action, while aminoglycosides, vancomycin, clindamycin, furosemide, beta blockers, calcium channel blockers and corticosteroids may enhance or prolong their action), age (alternations in drug elimination), hypothermia, sepsis, electrolyte concentrations and hepatic and renal function [22,31,32]. Tachyphylaxis has been also reported, especially in patients on continuous infusion; a change in NMBA class should be considered in such cases when the need for neuromuscular blockade is extended [33]. 

## 3. Rationale for NMBA Use in ARDS

In ARDS, the adoption of a lung protective ventilator strategy may diminish the risk of ventilator-induced lung injury and is associated with improved survival [9,10]. The use of NMBAs may facilitate lung protective ventilation, preventing spontaneous respiratory activity, thus limiting the risk of generation of large transpulmonary pressure swings, when strong inspiratory efforts occur. Moreover, expiratory efforts may lead to loss of aeration in dependent lung regions (de-recruitment), if pleural pressure is higher than the applied PEEP [34]. As a consequence, NMBAs may control tidal volumes and PEEP throughout the airways, minimizing the risk of barotrauma, volutrauma and atelectrauma. Finally, abolishing spontaneous efforts, NMBAs harmonize the patient–ventilator interaction. Vigorous spontaneous respiratory efforts may increase global transpulmonary pressures and tidal volumes and cause lung overdistension [35]. The deleterious effects of increased lung stress may also be present at a local level, involving certain lung regions, especially in dependent lung zones when the Pendelluft phenomenon occurs—the redistribution of air in lung regions from adjacent alveoli, causing local overdistension [36]. Moreover, some forms of asynchrony, such as the double triggering–delivery of a second tidal volume before complete exhalation, may result in the delivery of higher than set tidal volumes, increasing the local lung stress in the already-injured lung units [37]. As a result, increased respiratory drive, due to the underlying disease or triggered by increased PaCO_2_ (permissive hypercapnia), which may aggravate lung injury, is blunted with the use of NMBAs. It should also be mentioned that negative intrathoracic pressures may increase the intrathoracic blood volume, increasing lung perfusion; in the already-injured lung parenchyma, the capillary endothelium is already affected (increased permeability). Thus, additional lung damage occurs from the so-called negative pressure pulmonary edema [38].

Studies with daily interruption of sedation showed that patients had a shorter duration of mechanical ventilation, ICU stay and at least no negative effect on mortality [39,40]. These trials did not exclusively include ARDS patients. On the other hand, in ARDS patients, the decreasing work of breathing decreases the oxygen consumption by the respiratory muscles. It has been found that muscular paralysis results in the decrease in cardiac output and whole-body oxygen consumption, thus it has been speculated that blood flow is redistributed from the respiratory muscles to other vascular beds [41]. Finally, an anti-inflammatory role of NMBAs has been proposed. In patients with moderate and severe ARDS, the early use of muscular relaxants was associated with decreased concentrations of pulmonary and systemic proinflammatory markers, namely IL-1β, IL-6 and IL-8 [42]. Sottile et al., in a secondary analysis of the ARMA study, demonstrated that, in patients with PaO_2_/FiO_2_ < 120 mmHg receiving low tidal volume ventilation, a reduction in markers representing endothelial and epithelial lung injury was noted for each day of NMBA use [43]. The effects could be attributed to the decrease in lung inflammation, translated in a reduction in biotrauma, resulting from the optimization of patient–ventilator synchrony, or to a direct anti-inflammatory effect of myorelaxants, as shown in animal studies [44].

## 4. Monitoring 

The adequacy of NMBAs may be measured through the stimulation of peripheral nerves that produce twitches in the corresponding muscles. Train-of-four (TOF) monitoring involves a series of four electrical impulses (2 Hz) delivered to a peripheral nerve (more often the ulnar nerve), which results in the contraction of the corresponding muscle group. The amplitude of the twitches can provide information on the percentage of the AChRs that are occupied. The presence of fourth twitch indicates that 0–5% of the receptors are occupied; if there are only three twitches, 65–75% of the receptors are occupied and when there is no contraction, 100% paralysis is present [45]. Using the TOF monitoring, even when titrated to have a zero response out of four impulses (TOF 0/4), a nurse-driven protocol for NMBA titration was able to decrease the cisatracurium cumulative dose administered in 30 ARDS patients, without affecting the quality of the neuromuscular block [46]. It should be pointed that the dose achieved was less than half the dose used in the ACURASYS trial (14 ± 4 vs. 37.5 mg/h, *p* < 0.001) [21,46]. On the other hand, a randomized trial comparing cisatracurium dosing titration with TOF vs. clinical assessment concluded that careful titration of the neuromuscular blocking agent by clinical assessment alone is sufficient in patients undergoing continuous cisatracurium neuromuscular blockade [47]. Thus, current guidelines suggest that clinicians should use the strategy with which they are more familiar [48]. 

## 5. Potential Adverse Effects

One of the major problems when administering NMBAs is the possibility of increased risk of ICU-acquired weakness (ICUAW) and critical illness myopathy/polyneuropathy, which is present in about one in four ICU patients [49]. Well-recognized risk factors are prolonged ICU stay and mechanical ventilation, while neuromuscular blockade use was expected to increase patients’ vulnerability [50]. However, a meta-analysis of the published randomized controlled trials (RCTs) showed that NMBA use was not associated with increased risk of ICU-acquired weakness, and there was no significant heterogeneity between trials (Relative Risk (RR) 1.23; 95% CI, 0.99 to 1.53; *p* = 0.06; I^2^ = 0%) [51]. Of course, one should keep in mind the duration of NMBA administration. In all studies, NMBAs were administered for up to 48 h [51]. During the SARS-CoV-2 pandemic, NMBAs were administered for longer periods (even more that 5 days) [52,53]; a longer duration was not associated with extubation failure, although patients receiving NMBAs for a median of 8 days had a worse survival rate than patients receiving the agent for a median of 5 days [52,53]. The patients with a need for longer NMBA use had a significantly higher SOFA score upon the initiation of mechanical ventilation and worse respiratory system mechanics (driving pressure) [53]. In a cohort of 70 survivors with COVID-19 ARDS patients, 74% of the patients had received NMBAs for over 48 h. ICUAW was present in 66% of the patients and a longer NMBA duration was an independent risk factor for its occurrence [54]. Apart from the duration, the class of NMBA could also be implicated in the ICUAW, due to the incorporation of a steroid moiety in the aminosteroidal class, increasing the risk for muscle weakness. In fact, all randomized controlled trials that assessed ICUAW used cisatracurium [20,21,27,34,42,55,56]. Only Lyu et al. and Rao et al., two Chinese RCTs, used vecuronium (belonging to the aminosteroid class), which did not report on muscle weakness [27,56]. 

It is believed that concomitant corticosteroid use increases the vulnerability to ICYAW. In an ACURASYS study, 39.5% of the patients received corticosteroids; however, there was no increased incidence of ICUAW in the NMBA group [21]. Moreover, in COVID-19 patients, the longer duration of the agents’ administration and not corticosteroid use was identified as a risk factor [54]. It must be taken into account that hepatic or renal failure may also decrease the elimination of aminosteroid NMBAs and prolong their action [22]. Another study pointed out that the duration of bed rest, rather than the cumulative steroid or NMBA dose, was responsible for muscle weakness [57]. 

Complications that should also be monitored are deep venous thromboembolism (DVT); Boddi et al. reported that NMBA treatment was the strongest factor influencing DVT incidence [58]. Impaired eyelid closure is a rather important issue, as the cornea is at the risk of drying, scarring, ulceration, infection and finally visual loss [22]. Awareness with paralysis should not be neglected when NMBAs are used. Muscle relaxants do not provide analgesia or sedation and an assessment of adequate analgosedation should be assessed. This is considered as a “good practice statement” in the guidelines from the Society of Critical Care Medicine, preventing traumatic recollections, negative experiences, dreams or thoughts [33,59]. Monitoring awareness with bispectral index has been found inadequate, thus the careful clinical monitoring of patients’ signs and symptoms is suggested, while a brief NMBA discontinuation could also be performed [60,61]. 

## 6. Data on NMBA Use in ARDS

Seven RCTs have been conducted evaluating the benefits, if any, of NMBA use in ARDS patients [20,21,27,34,42,55,56]. The earliest four studies were performed in France [21,34,42,55], two subsequent studies in China [27,56] and the latest, also including the larger number of patients, was performed in USA [20]. The studies have provided conflicting results, so that they have changed the clinical practice in a bidirectional way. Especially when considering the most influential ones (ACURASYS and ROSE study) with the highest recruitment, certain differences have been addressed, although the design of the ROSE study was carefully selected to allow direct comparisons to ACURASYS [20,21].

In the first study in the field, Gainnier et al. randomized 56 patients with moderate and severe ARDS within 36 h of the patients meeting the eligibility criteria. The patients assigned to the NMBAs group received cisatracurium at a dose of 5 μg/kg/min (cumulative dose of 1324 ± 197 mg) after a bolus infusion of 50 mg. The study found that the early use of NMBAs resulted in sustained improvements in oxygenation after 48 h of infusion, persisting during the 120 h of the study period. Hospital mortality on day 28th, 60th and ICU mortality did not differ (Table 1).

Only one patient suffered a pneumothorax in the control group and none of the patients developed any signs of muscle weakness, while there were no differences in the number of patients suffering a ventilator-associated pneumonia (VAP) episode. The second RCT randomized 36 patients with ARDS and PaO_2_/FiO_2_ < 200 mmHg to receive NMBAs for 48 h or not. The authors reported that the sustained improvement in oxygenation was accompanied with a reduction in pulmonary and systemic cytokines in patients under NMBAs. Ventilator-free days (VFDs) and ICU mortality (27.8% vs. 55.6%) did not differ (Table 1) [42]. The small sample size may have precluded the numerical difference in mortality to reach a statistical significance. The same study group reported that, in 30 ARDS patients with moderate and severe ARDS, the use of NMBAs increased the mean inspiratory and expiratory transpulmonary pressures, as a result of a stable positive end expiratory pressure with the elimination of expiratory efforts [34]. The changes in lung mechanics were associated with improvements in oxygenation in the NMBA group. The authors point that, by abolishing expiratory efforts, a significant amount of derecruitment during expiration can be achieved [34]. There are two Chinese RCTs using vecuronium as the NMBA: Lyu et al. randomized 96 patients with moderate (48 patients) and severe ARDS (48 patients). The patients with PaO_2_/FiO_2_ < 100 mmHg receiving 24–48 h vecuronium presented significant improvements in oxygenation, perfusion and multiple severity scores, while they had also lower 21-day mortality rates (20.8 vs. 50%, *p* = 0.04) [27]. Both studies did not report significant adverse effects with the use of aminosteroidal NMBAs. 

The most robust evidence concerning the use of NMBAs in ARDS patients comes from the two largest RCTs (ACURASYS and ROSE studies), although reporting conflicting results [20,21]. The ACURASYS study was a multicenter RCT, conducted in France, which randomized 340 patients to receive a 48 h of cisatracurium infusion or placebo, within 48 h of ARDS diagnosis [21]. The primary outcome was 90-day mortality. The study team followed the same NMBA administration protocol as the previous studies [34,42,55]. Using a continuous infusion of 37.5 mg (after a bolus of 15 mg) of cisatracurium, as it had been found adequate to sustain paralysis in the previous studies, no peripheral nerve simulation was performed. The authors found that the hazard ratio for death at 90 days in the cisatracurium group compared to the placebo group was 0.68 (95% confidence interval (CI), 0.48 to 0.98; *p* = 0.04) after adjustment for the baseline PaO_2_/FiO_2_, SAPS II and plateau pressures. The crude 90-day mortality was 31.6% (95% CI, 25.2 to 38.8) in the cisatracurium group and 40.7% (95% CI, 33.5 to 48.4) in the placebo group (*p* = 0.08). Patients in the intervention group had also more VFD in the first 28 days (10.6 ± 9.7 vs. 8.5 ± 9.4, *p* = 0.04) and an increased hazard ratio for weaning from mechanical ventilation by day 90 (HR 1.41 95% CI 1.08 to 1.83; *p* = 0.01). Barotrauma was significantly more frequent in the placebo group (4 vs. 11.7%, *p* = 0.01). 

In 2014, the National Heart, Lung and Blood Institute (NHLBI) launched the Prevention and Early Treatment of Acute Lung Injury (PETAL) Network to conduct phase III trials to test treatments with the potential to improve clinical outcomes of patients with or at risk of developing ARDS. PETAL built on a new network (NHLBI ARDS Clinical Trial Network (ARDSNet)) with a focus on early treatment and prevention. Thus, in 2016, PETAL launched the Re-evaluation of Systemic Early Neuromuscular Blockade (ROSE) trial to assess the efficacy and safety of early neuromuscular blockade in reducing mortality and morbidity patients with moderate and severe ARDS [63]. The outcomes were tested on a longer timeframe (up to 12 months). The rational for the conduction of a second trial is summarized below: Firstly, there was a need to re-evaluate the effectiveness of NMBAs in a larger cohort than the one included in ACURASYS study [21]. In view of the change in clinical practice favoring light sedation [39,40], the ROSE protocol intended to compare heavy sedation with additional paralysis with NMBAs to a light sedation protocol adopted in the control group. Thirdly, the patients were randomized if they presented a PaO_2_/FiO_2_ <150 mmHg with PEEP application of 8 cm H_2_O or higher. This is in contrast to the inclusion criterion used in the ACURASYS trial, where hypoxemia was tested with a PEEP of at least 5 cm H_2_O [21]. This criterion was selected to exclude patients with transient hypoxemia after intubation [63]. The trial enrolled 1006 patients and was stopped for futility after the second interim analysis; the decision was independently made considering the data analyzed and the safety monitoring results. Treatment with NMBAs was not associated with increases in oxygenation variables, nor were there any effects on in hospital mortality or VFM. The incidence of barotrauma did not differ across the patients in both study arms, while patients receiving muscular relaxants presented serious cardiovascular adverse effects (one death from complete heart block and refractory shock) [20]. There are certain differences that may explain the contradictory results in these two RCTs. Firstly, the approach concerning the ARDS treatment was quite different. The amount of PEEP used in ROSE trial was much higher than the one used in the ACURASYS study (12.6 ± 3.6 vs. 9.2 ± 3.2 cm H_2_O), prone positioning was less frequently used (16% vs. 29%), while the patient enrollment in the ROSE study was too quick (actual time of enrollment was 8 h from meeting eligibility criteria). Some patients might have improved in the next few hours only with the application of the ventilator (i.e., PEEP application) and not be eligible for the study. The profound difference, though, between the two studies is the light sedation protocol adopted in patients with moderate and severe ARDS. With the results of both trials, the most recent publication for the guidance on neuromuscular use in ARDS patients highlights that patients with moderate or severe ARDS should not receive heavy sedation and NMBAs if they can be ventilated with light sedation (absence of rigorous respiratory efforts, high esophageal pressure swing, increased respiratory rate and/or voluntary expiration) [48].

## 7. Subsequent Studies

### 7.1. Pediatric Population

Recently, in a secondary analysis of data from the Randomized Evaluation of Sedation Titration for Respiratory Failure (RESTORE) clinical trial, a pediatric multicenter cluster randomized trial of sedation, found that the early use of NMBA was associated with a longer duration of IMV (Hazard Ratio (HR), 0.60; 95% CI 0.5–0.72; *p* < 0.0001), but, after adjusting for confounders, the 90-day in-hospital mortality did not differ (OR, 1.92; 95% CI, 0.99–3.73; *p* = 0.053) nor did extubation failure (7% vs. 8%, *p* = 0.45). Functional or cognitive impairment at hospital discharge was also comparable between pediatric patients receiving early NMBAs and those receiving late or not at all NMBAs [64].

### 7.2. COVID-19 ARDS Experience

During the last almost three years of SARS-CoV-2 pandemic, a substantial number of ARDS patients was admitted in the ICUs, overwhelming ICU and physicians’ “capacity”. There was a large debate concerning the ventilatory parameters that should be used, especially concerning the PEEP level, as many intensivists evaluated physiology at the bedside, following the published data on driving pressure and PEEP [10,11]. Concerning NMBA use, in the initial guidance, there was a weak recommendation to use NMBAs for up to 48 h in the event of persistent ventilator dyssynchrony, the need for ongoing deep sedation, prone ventilation or persistently high plateau pressures [62]. Since then, although thousands of patients have been treated for SARS-CoV-2 ARDS, there is a wide variance in the reported use; on the contrary, there is only scarce information on NMBA duration. It seems the clinical practice during the pandemic extends beyond the recommendations [63,64].

In a European multicenter report, NMBAs were administered in 72% of the COVID-19 ARDS patients. Surprisingly, 64% of the patients with mild ARDS were also receiving the treatment, although the duration is not reported [65]. The largest report on 4244 patients pointed that NMBAs were used in 82% of the patients in Europe [66]. It seems that the need for NMBAs has increased, as only 24% of the patients included in the PRoVENT-COVID study, covering the first epidemic wave, had received myorelaxants with a median duration of 8 h [67]. In France, during the same period, NMBAs were administered in 48% of the patients admitted in the ICU with a median PaO_2_/FiO_2_ of 112 mmHg [68]. Almost half of the patients were reported to receive a blocking agent in other studies as well [69,70,71]. On the contrary, Dres et al., in a multicenter study including 149 ICUs, NMBAs were used in 86% of the intubated patients [72].

Initial reports on patient characteristics were only limited in the NMBA use. Only few studies report on the duration of use. Courcelle et al., in an observational, multicenter study, evaluated 407 COVID-19 ARDS patients with PaO_2_/FiO_2_ < 150 mmHg aiming to describe the practice and association of NMBA use with 28-day outcomes [52]. They found that NMBAs were used in 84% of the patients. On the contrary, the clinical practice highlighted in the LUNGSAFE study indicated the use of these agents in 24% of the patients [3]. There were 241 (59%) COVID-19 ARDS patients treated with NMBAs for more than 48 h. The median duration of NMBA use was 5 days (interquartile range 2–10 days) [52]. The authors performed a propensity matched analysis in 103 patients that showed no difference in clinical characteristics and 28-day outcomes (ventilator-free days, day 14 mortality and length of hospital stay) between the patients receiving NMBAs for more than 48 h (Table 1). In another study, NMBA use favored survival in 129 patients with moderate and severe COVID-19 ARDS [53]. Ninety-eight (76%) of the patients received an NMBA and the need for the treatment use was also prolonged (median duration 5 days—IQR 4–9). Survivors were using NMBAs more frequently than non-survivors (84% vs. 63%, *p* = 0.006), while in the latter group, a longer period of myorelaxant use was observed (median duration 8 compared to 5 days in survivors, *p* = 0.008). Non-survivors had higher SOFA scores upon MV initiation and worse lung mechanics. Li Bassi et al. extracted data from the multicenter registry of the international COVID-19 Critical Care Consortium, incorporating the Extra Corporeal Membrane Oxygenation for 2019 novel Coronavirus Acute Respiratory Disease (COVID-19-CCC/ECMOCARD), to delineate the role of NMBAs in ventilated COVID-19 patients [73]. It is surprising to find that, contrary to other reports, among the 1953 COVID-19 ARDS patients under invasive mechanical ventilation, only 12.39% are reported to receive an NMBA. The study period extended from February 2020 to October 2021. Among the NMBA cohort, 74.4% received myorelaxants for 48 h and the rest for 3 days. In the propensity analysis, NMBA use had no effect on 90-day mortality (unadjusted hazard ratio 1.12, 95% CI 0.79 to 1.59, *p* = 0.534). In a sensitivity analysis, matching the patients for smoking, use of antibiotics, antivirals, corticosteroids, renal replacement therapy, ECMO and prone positioning, continuous NMBA use beyond the third day showed an increased risk for 90-day mortality (adjusted HR 1.73, 95% CI 1.22, 2.37). Yet, the results of the study should be interpreted with caution as the median duration of mechanical ventilation in the patients not receiving NMBA was only 2 days. 

A Spanish report on the incidence of ICUAW in a cohort of 70 patients surviving after treatment for severe COVID-19 ARDS indicated that, in patients with such severe impairment in gas exchange, NMBAs were used in a total of 87.1% of the patients and in 74.3% the duration was extended beyond the first 48 h [54]. ICUAW was present in 65.7% upon ICU discharge and persisted in 31.4% at hospital discharge. The occurrence of ICUAW was associated with the use and duration of NMBAs, the longer need for invasive mechanical ventilation and not with the use of corticosteroids. Rodriguez et al. evaluated 31 COVID-19 ARDS patients for critical illness myopathy presence. Seventeen patients developed myopathy and among treatments; NMBAs were administered for half of the total ICU stay (median duration of ICU stay of 19 days) [74]. Ego et al. compared the sedative strategies used in 39 COVID-19 ARDS and 39 non-COVID-19 ARDS patients. NMBAs were administered in all the patients for a median duration of 12 vs. 3 days in the respective groups (*p* < 0.001) [75]. The abovementioned studies indicate that NMBAs are frequently used in the management of COVID-19 ARDS, although the plateau pressures are frequently within the acceptable limits. The increased respiratory drive seems to be one of the major issues mandating the use of NMBAs in conjunction with sedation in COVID-19 ARDS [76]. Vigorous breathing efforts may amplify the severity of lung injury, which in turn can influence the duration of mechanical ventilation and impact patient outcome [77]. Apart from reflex stimulation from the injured lungs, COVID-19 may affect angiotensin-mediated sensitivity of the carotid bodies (which express ACE2 receptors) and generate more complex brainstem-level alterations of the control of breathing, regardless of the degree of hypoxia or lung mechanics. These complex interactions may be amplified over time from imputes arising from changes in lung mechanics, ventilatory needs and neural transmissions [77].

## 8. Conclusions

The management of ARDS patients should focus on minimizing the exacerbation of lung injury during the need for application of artificial ventilation. Therefore, patients should be treated with the least mechanical power applied to the lungs that can be accomplished. Hence, a low tidal volume, PEEP titrated to lung mechanics (driving pressure) and the least respiratory rate that can be tolerated should be applied. Sedation should be titrated to facilitate lung-protective ventilation. If all the above cannot be accomplished due to increased respiratory drive or patient ventilatory dyssynchrony, NMBAs should be applied. Daily assessments to discontinue the myorelaxants and encourage spontaneous breathing activity is a logical approach. If NMBAs are needed to facilitate lung-protective ventilation, a partial neuromuscular blockade is an attractive option that should be evaluated in future studies, especially in COVID-19 ARDS patients with increased ventilatory drive. 

## Figures and Tables

**Table 1 jpm-12-01538-t001:** Patient characteristics and outcomes in the seven randomized controlled trials and the COVID-19 ARDS studies concerning NMBA use in ARDS.

	Patients	Primary End Point	Time of Inclusion	ΝΜΒA	Dose	Monitoring	Duration	Sedation	PEEPtot	VT	Pplat	PaO_2_/FiO_2_	Proning	Steroids	Barotrauma	VAP	ICUAW	VFD 28/60	Mortality
Gainnier [55]	56	Effect on oxygenation after 120 h	Within 36 h meeting inclusion criteria	cis	50 mg bolus	TOF every 8 h	48	midazolam sufentanil	12.3 ± 3	7.1 ± 1.1	27.1	130 ± 34	14.3%	7.1%	0	46%	0	D28: 3.7 ± 37.2	D28: 35.7%
2004	PaO_2_/FiO_2_ < 150 mmHg				5 μg/kg/min	Ramsey 6												D60: 19 ± 320.3	D60: 46.4%
																			ICU: 46.4%
					actual: 1324 ± 197				11.4 ± 2.5	7.4 ± 31.9	26.1 ± 4	119 ± 31	14.3%	14.3%	1	57%	0	D28: 1.7 ± 35.3 (NS)	D28: 60.7 (*p* = 0.061)
																		D60: 9.8 ± 16.9 (0.071)	D60: 64.3% (*p* = 0.18)
																			ICU: 71.4% (*p* = 0.057)
Forel [42]														5.5%					
2006	36	Effect on pulmonary and systemic proinflammatory cytokines	48 h of ARDS onset	cis	bolus 0.2 mg/kg	TOF every 8 h	48 h		13.2 ± 2.7	6.5 ± 0.7	27.5 ± 4.4		0		0		1	D28: 6 ± 8.6	ICU: 27.8%
	PaO_2_/FiO_2_ < 200 mmHg				5 μg/kg/min				11 ± 2.7 (*p* < 0.05)	7 ± 30.7	24.8 ± 35.7			0	0		1	D28: 5.4 ± 6.4 (ns)	ICU: 55.6% (NS)
Guervilli [34]	30	Effect on respiratory mechanics	48 h of ARDS onset	cis	bolus 15 mg	TOF	48 h	midazolam/sufentanil	11 (10–11.5)	6.2 (5.9–6.8	23 (19–26)	158 (131–185)						D28: 7 (0–20)	ICU: 38%
2017	PaO_2_/FiO_2_ 100–150 mmHg				37.5 mg/h	Ramsey 6						150 (121–187)						D28: 8(0–18)	ICU: 27% (*p* = 0.6)
Rao [56]	41			vecuronium	actual: 1595 mg (1221–1830)				7.84 ± 2.94							4.2%		D28: 17.9 ± 2.77.4	D28: 4.2%
2016	ARDS pts																		90: 4.2%
																0		D28: 17.1 ± 8.2	D28: 11.8%
									5.88 ± 1.96										90: 17.6%
Lyu [27] 2014	96			vecuronium	0.05 mg/kg/h		24–48 h					140.95 ± 26.97							D21: 16.7%
	PaO_2_/FiO_2_ < 200 mmHg (24)																		D21: 25% (*p* = 0.035)
	PaO_2_/FiO_2_ < 100 mmHg (24)											144.33 ± 24.09							D21: 20.8%
																			D21: 50% (0.477)
Papazian [21]	340		48 h of ARDS onset	cis	bolus 15 mg	Ramsay 6		midazolame	9.2 ± 3.2	6.55 ± 1.12	25 ± 5.1	106 ± 36	28%	16%	4%		28D: 70.8%	28D: 10.6 ± 9.7	28D: 23.7%
2010	PaO_2_/FiO_2_ < 150 mmHg	90-day mortality	actual inclusion time: 16 h		37.5 mg/h			sufentanil									icu discharge: 64.3%	90D: 53.1 ± 35.8	ICU: 29.4%
								ketamin											In hospital: 32.2%
								propofole											
									9.2 ± 3.5	6.48 ± 0.92	24.4 ± 4.7	115 ± 41	29%	23%(*p* = 0.1)	11.7% (*p* = 0.01)		28D:67.5% (*p* = 0.64)	28D: 8.5 ± 9.4 (*p* = 0.04)	28D: 33.3% (*p* = 0.05)
																	icu discharge: 68.5% (*p* = 0.51)	90D: 44.6 ± 37.5 (*p* = 0.03)	ICU: 38.9% (*p* = 0.06)
																			In hospital: 32.2% (*p* = 0.08)
ROSE [20]	1006	90-day mortality	48 h of ARDS onset	cis	bolus: 15 mg	Ramsey 5–6	48 h		12.6 ± 3.6	6.3 ± 0.9	25.5 ± 6.0	98.7 ± 27.9	16.8%	17%	4%		D7: 41%	D28: 9.6 ± 10.4	D90: 42.5%
2019	PaO_2_/FiO_2_ < 150 mmHg		actual randomization time: 8 h		37.5 mg continuous infusion	RASS −5											D28:46.8%		D28: 36.7%
						Ramsey 2–3			12.5 ± 3.5	6.3 ± 0.9	25.7 ± 6.1	99.5 ± 27.9	14.9%	16.4%	6.3%		D7: 31.3%	D28:9.9 ± 10.9	D90: 42.8%
						RASS 0 to −1											D28:27.5%		D28: 37%
Courcelle [52]	407	NMBA use					5 days (IQR 2–10)		<48 h: 12 (10–14)	6.1 (5.8–6.6)	23 (20–26)	126 (88–162)	65%					D28: 0 (0–16)	ICU: 38%
2020	PaO_2_/FiO_2_ < 150 mmHg	28-day outcomes											Propensity cohort 78%						
	COVID-19 ARDS																		
									>48 h: 11 (10–13)	6.1 (5.8–6.6)	24 (21–26)	120 (87–157)	90% (*p* < 0.001)					D28: 0 (0–10)	ICU: 41% (*p* = 0.54)
													propensity cohort: 80% (*p* = 0.86)						
Lee [53]	129	ICU mortality					5 days (4–9)		survivors: 10 (9–12)	7 (6.2–7.9)		123 (87–197)	16%	92%		53% (superinfection rate)		8.2 ± 9.7	ICU 37%
2022	COVID-19 ARDS												survivors: 20%	91%					mild ARDS: 20%
													non-survivors: 10%	94%					moderate ARDS: 40%
									non-survivors: 10 (10–12)	6.8 (6.2–8.3)		109 (85–134)							severe ARDS: 43%
Li Bassi [62]																			
2022	1953	90-day mortality							No NMBA: 12 ± 3	7.1 ± 1.4	25.4 ± 5.7	98.1 ± 31.1	8.6%	21.3%	12.4%				
	COVID-19 ARDS (moderate and severe)								No NMBA(PS): 11.9 ± 2.73.1	7.4 ± 1.6	25 ± 2.75.9	86 ± 30.7	10.5%	22.9%	9.6%				
	242 with early NMBA						48 h: 74.4%		NMBA: 12.8 ± 3.3	6.9 ± 1.4	26.1 ± 2.75.1	88.5 ± 29.3	21.5%	19.8%	9.6%				
							72 h: 25.6%		NMBA (PS): 12.8 (3.3)	6.8 ± 1.4	26.2 ± 2.75	88.6 ± 29.7	21.9%	19%	10.4%				
Nunez-Seisderos [54]	70 survivors with COVID-19 ARDS	ICUAW		cis	cumulative dose: 739 mg (283–1425)		5 days (2–8)					81 (64–97.75)	91.4%	100%			65.7%	IMV DUR: 13 (7–22.5)	
2022																			

ARDS: Acute Respiratory Distress Syndrome; ICU: Intensive Care Unit; ICUAW: Intensive Care Unit-Acquired Weakness; IMV DUR: Invasive Mechanical Ventilation Duration; NMBAs: Neuromuscular Blocking Agents; PEEP: Positive-End Expiratory Pressure; TOF: Train of Four; VFD: Ventilator-Free Days.

## Data Availability

Not applicable.

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
