# Peer review of "Neuromuscular Blockade in the Pre- and COVID-19 ARDS Patients"

_jpm, 2022, doi:10.3390/jpm12091538_

Round 1
Reviewer 1 Report
The manuscript does excellent job demonstrating NMBAs use in the pre- and Covid-19 Ards patients. A few typo must be fixed.
Author Response
Comment 1: The manuscript does excellent job demonstrating NMBAs use in the pre- and Covid-19 ARDS patients.
Reply 1: We want to thank the reviewer for the valuable suggestion and encouraging comments.
C1: A few typo must be fixed.
R1: Done as recommended.

Reviewer 2 Report
Very good summary of pre-COVID and COVID-associated findings. If possible, suggest conducting a meta-analysis of the trials from these two periods. Understandably, there will be more case series and non-randomized reports.
Author Response
Comment: Very good summary of pre-COVID and COVID-associated findings. If possible, suggest conducting a meta-analysis of the trials from these two periods. Understandably, there will be more case series and non-randomized reports.
Reply : We want to thank the reviewer for the valuable suggestion and encouraging comments. According to your recommendations we reviewed the literature and have added 10 more references concerning the incidence of neuromuscular blocker use in Covid-19 ARDS patients (Ref 65-74). Of these only Ref 73 and Ref 74 report on the duration of NMBA use. Please find the revised data in lines 348-360 and lines 397-403.
